# The Formation and Renewal of Photoreceptor Outer Segments

**DOI:** 10.3390/cells13161357

**Published:** 2024-08-15

**Authors:** Jingjin Xu, Chengtian Zhao, Yunsi Kang

**Affiliations:** 1College of Marine Life Sciences, Ocean University of China, Qingdao 266003, China; 13969737352@163.com (J.X.); chengtian_zhao@ouc.edu.cn (C.Z.); 2MoE Key Laboratory of Evolution and Marine Biodiversity, Institute of Evolution and Marine Biodiversity, Ocean University of China, Qingdao 266003, China

**Keywords:** photoreceptor, outer segments renewal, retinitis pigmentosa, phagosomes, retinal diseases

## Abstract

The visual system is essential for humans to perceive the environment. In the retina, rod and cone photoreceptor neurons are the initial sites where vision forms. The apical region of both cone and rod photoreceptors contains a light-sensing organelle known as the outer segment (OS), which houses tens of thousands of light-sensitive opsins. The OSs of photoreceptors are not static; they require rhythmic renewal to maintain normal physiological functions. Disruptions in OS renewal can lead to various genetic disorders, such as retinitis pigmentosa (RP). Understanding the patterns and molecular mechanisms of photoreceptor OS renewal remains one of the most intriguing topics in visual biology. This review aims to elucidate the structure of photoreceptor OSs, the molecular mechanisms underlying photoreceptor OS renewal, and the retinal diseases resulting from defects in this renewal process. Additionally, we will explore retinal diseases related to photoreceptor OS renewal and potential therapeutic strategies, concluding with a discussion on future research directions for OS renewal.

## 1. Introduction

Photoreceptors are sophisticated sensory neurons that generate electrical responses when stimulated by light [1]. All processes responsible for photon capture and the generation of electrical signals occur in the photoreceptor OS [2]. The OS comprises thousands of stacked membrane discs, a specialized structure designed to maximize visual sensitivity and enhance phototransduction efficiency. During the ongoing process of phototransduction, a significant amount of proteins and lipids within the OS of photoreceptor cells undergo oxidation [3]. As a result, the photoreceptor OS needs continuous renewal to maintain its function. The old OSs at the apexes of photoreceptor cells are phagocytosed by retinal pigment epithelium (RPE) cells, while the inner segment (IS) synthesizes the opsins and other membrane-associated proteins necessary for the OS discs. These proteins are transported to the OS via the connecting cilia, where they form new discs [4,5,6,7,8]. Disruptions in OS renewal can lead to various genetic disorders, such as RP. The renewal of photoreceptor OSs has become an increasingly important topic in ophthalmology research. With the advancements in molecular biology and imaging technologies, the patterns and molecular mechanisms of photoreceptor OS renewal are poised to be studied more comprehensively.

In this review, we will first discuss the structure of OSs and the molecular mechanisms of OS formation, emphasizing the differences between rod and cone OSs. Next, we will review early discoveries on the cyclical nature of photoreceptor OS renewal and the circadian rhythms involved in the renewal process. Subsequently, we will introduce recent advancements in the molecular mechanisms of OS renewal, related retinal diseases, and potential therapeutic strategies. Finally, we will address some remaining issues in this field of research.

## 2. Photoreceptor OS

The photoreceptor OS is a highly specialized primary sensory cilium [2]. Unlike other cilia, this cilium is composed of tightly stacked membrane discs containing visual pigments, a structure essential for efficient light capture [1].

### 2.1. Structure and Differences between Rod and Cone OSs

Rod and cone photoreceptor OSs exhibit distinct structural organization and three-dimensional shapes. Rod OSs are elongated, whereas cone OSs are conical in shape. Rod OSs are longer compared to cone OSs, presenting a notable contrast in length. The tapering of cone OSs is particularly pronounced in lower vertebrates [9] (Figure 1A,B).

In addition to morphological differences, the most significant distinction between rod and cone photoreceptors lies in the structure of their membrane discs. In rod cells, the OS is divided into two separate membrane domains: the discs and the plasma membrane. The mature discs are completely separated from the plasma membrane, a configuration referred to as “closed discs” [10] (Figure 1A). In contrast, most cone OS discs are continuous with the plasma membrane, known as “open discs” [11] (Figure 1A). However, this description is not absolute. Mammalian cone OSs can also contain discs that are separated from the plasma membrane. In primate cone cells, the basal one third of the cone OS discs, or sometimes more, are found to be continuous with the plasma membrane, while the distal region of the cone OS features closed discs [12,13,14]. In mammalian rods, only a few discs at the base of the OS are continuous with the plasma membrane, with the rest being fully internalized as closed discs [15,16,17,18,19]. Overall, most rod discs are entirely internalized, whereas most cone discs maintain continuity with the plasma membrane.

The connecting cilium is the main structural component of photoreceptor OSs and shares similarities in structure with the primary cilium. Situated at the base of both the connecting cilium and the primary cilium is the basal body, functioning as the central organizing hub for microtubules (Figure 1A,C) [20]. The basal body contains nine triplet microtubules, two of which extend to form the axoneme, while the third terminates early to anchor the basal body to the plasma membrane through transitional fibers [21]. The microtubule doublets gradually transition to singlets at the distal end of the axoneme [22] (Figure 1A). Across most species, the axoneme typically encompasses only one-third of the rod OS’s length [21,23]. Conversely, in cone OSs, the axoneme extends along the entire length of the OS [1,24] (Figure 1A).

### 2.2. Morphogenesis and Molecular Basis of Photoreceptor OSs

#### 2.2.1. Morphogenesis of Photoreceptor OS Discs

Photoreceptor OSs play a crucial role in light perception, which is intricately linked to their precise structure. Why do thousands of membranous discs align in a parallel, orderly fashion within the OS? There used to be two hypothesized models for the morphogenesis of rod OS discs: the vesicle fusion model and the plasma membrane evagination model [25] (Figure 1D).

The vesicle fusion model posits that the discs are formed by the fusion of vesicles and tubular compartments at the base of the OS. These vesicles may originate from the plasma membrane of the OS or from transport vesicles that move through the connecting cilium. New discs are formed through vesicle fusion at the base of the OS, followed by a flattening process that shapes them into mature discs. Supporting this hypothesis is the observation of vesicular structures at the base of rod OSs in mice photoreceptors [7,26] (Figure 1D).

The plasma membrane evagination model, proposed by Steinberg, suggests that membrane discs originate from the evagination of the plasma membrane at the base of the OS [27]. Initial evidence supporting this model arose from transmission electron microscopy observations of the basal region of rod OSs in macaque and frog photoreceptors, revealing the presence of open membrane discs [27,28]. Subsequently, refinement of the plasma membrane evagination model elucidated that the plasma membrane at the base of the OS undergoes evagination facilitated by filamentous actin (F-actin). Newly formed evaginated discs in photoreceptors maintain continuity with the plasma membrane [29,30]. These “U-shaped” open discs are referred to as “edges” [25] (Figure 1D). Following F-actin depolymerization, the newly formed evaginated discs undergo flattening, mediated by Progressive Rod-Cone Degeneration (PRCD) protein [31,32]. On the axonemal side, the newly evaginated discs develop curved “rims” due to the action of Retinal Degeneration Slow (RDS) protein [33,34]. As the plasma membrane continues to expand and inwardly invaginate away from the axoneme, older evaginated discs detach from the plasma membrane and become fully internalized as “closed” discs when the older membrane fuses with the rims of adjacent newly formed evaginated discs (Figure 1D) [25].

With the advancement of experimental techniques, three independent studies confirmed the membrane evagination model of disc morphogenesis [15,16,19]. None of these inquiries documented the presence of vesicles at the base of rod OSs. Moreover, they substantiated that the appearance of vesicles at the OS’s base is an artefact resulting from the disruption of membrane disc ultrastructure, attributed to delayed fixation or postmortem tissue manipulation [16].

#### 2.2.2. Connecting Cilia

The OS requires daily renewal, with proteins synthesized in the cell body being transported to the OS via the connecting cilia. This process necessitates a mechanism known as intraflagellar transport (IFT) [35]. Photoreceptor cells utilize IFT to transport cellular cargo from the cell body to the ciliary apical (anterograde transport mediated by Kinesin motor proteins) and from the ciliary apical back to the cell body (retrograde transport mediated by Dynein motor proteins). The IFT complex functions as a “large cargo container” carrying numerous macromolecular cargos [2]. In 1998, Cole discovered that the IFT complex is divided into two parts: IFT-A and IFT-B [36]. The absence of IFT components often disrupts ciliary elongation, leading to defects in photoreceptor OS development. The first evidence of IFT’s involvement in photoreceptor cilium formation and transport came from a hypomorphic mouse mutant of *Ift88^orpk^*. The mutant exhibited abnormal OS discs, accumulate rhodopsin in the IS, and the photoreceptors subsequently underwent apoptosis [37]. Similar results were observed in zebrafish photoreceptor cells lacking Ift88 [38,39]. Motor proteins are also crucial for photoreceptor development. Kinesin-II regulates anterograde transport in photoreceptors. It consists of two motor domains, KIF3A and either KIF3B or KIF3C [40]. Studies in mice have shown that the loss of KIF3A leads to photoreceptor degeneration and apoptosis [41]. In zebrafish, *kif3a^−/−^* mutants exhibit a similar phenotype. Interestingly, *kif3a^−/−^* mutants show more severe defects in both overall ciliary development and photoreceptor layer development compared to *kif3b^−/−^* mutants. This discrepancy may be attributed to functional redundancy between Kif3b and Kif3c [42].

#### 2.2.3. Essential Proteins Constitute the Structure of Photoreceptor OSs

The OS contains many unique proteins, and their distribution is highly ordered. Many of these proteins are essential for the formation and maintenance of the OSs. Here, we will discuss several structural proteins that are crucial for OS (Figure 1E).

Opsin, a member of the G-protein-coupled receptor (GPCR) family, is the primary component of the OS [43]. In rod OSs, rhodopsin constitutes approximately 90% of all OS proteins [44]. Rhodopsin is not only a crucial functional protein but also an essential structural component [45]. *Rho^−/−^* mice fail to develop normal rod OSs, and rod cells undergo degeneration and apoptosis within three months after birth, indicating that rhodopsin is important for rod cell survival [46]. Mutations in rhodopsin are the most common cause of RP in humans, accounting for approximately 25% of autosomal, dominantly inherited RP cases are caused by mutations in rhodopsin [47]. Recent reports have indicated that mice lacking cone opsin do not undergo apoptosis in cone cells, but they also fail to form normal OS structures [48]. Overall, opsins are the most critical protein components for the formation and maintenance of photoreceptor OS structures.

RDS, also known as Peripherin-2 (PRPH2), is a member of the tetraspanin (TSPAN) protein family. It is exclusively expressed at the rims of the disc membranes in rod and cone OSs (Figure 1C), specifically in the “hairpin-shaped” rims of the discs that are not connected to the plasma membrane and are fully internalized [49]. RDS directly induces membrane curvature, contributing to the formation of the hairpin rims during disc morphogenesis [34,50]. RDS is also necessary for the proper formation of both rod and cone photoreceptor OSs in mice [51]. *Rds^+/−^* mice exhibit abnormal OS development, shortened OSs, and whirl-like discs. Photoreceptor cells undergo slow degenerative apoptosis, with approximately 50% of the cells undergoing degeneration by the time the mice reach 18 months. Additionally, *Rds^+/−^* mice show larger phagosomes within the RPE, indicating impaired OS renewal in these mice [52]. *Rds^−/−^* mice cannot form OSs; photoreceptor cells begin to undergo slow apoptosis around 2 to 3 weeks after birth and nearly completely disappear within 1 year [52,53]. In humans, over 200 different RDS mutations have been linked to various inherited retinal diseases (IRDs) [54].

Retinal outer segment membrane protein 1 (ROM1) is also a member of the tetraspanin (TSPAN) protein family that can form a heterotetrameric complex with RDS [53,55]. ROM1 demonstrates a reduced ability to form non-covalent tetramers with RDS compared to the homotetramers formed by RDS alone [56]. ROM1 can also engage with RDS via disulfide bonds to form oligomers. However, unlike RDS, ROM1 exhibits rare independent oligomer formation [57]. In contrast to *Rds^−/−^
*mice, which completely fail to form OSs, *Rom1^−/−^* mice exhibit slight defects in rod cell OSs and slow photoreceptor degeneration [58]. Thus, ROM1 potentially modulates the distribution of RDS complex and the ratio of RDS to ROM1 [58].

Cyclic nucleotide-gated (CNG) channels are crucial for the visual transduction in the photoreceptors (rods and cones) of the vertebrate retina [59] (Figure 1E). Interestingly, components of the CNG channel proteins also contribute to OS maintenance. CNG channels are composed of CNGA and CNGB subunits, where the A subunit confers the primary channel characteristics, and the B subunit is essential for the localization of these complexes to the OS plasma membrane [59,60]. RDS can not only bind to the β subunit of the rod OS membrane CNG but also to two other non-membrane isoforms, GARP1 and GARP2 (CNGB1 splice-variant), known as glutamic acid-rich proteins [61]. Immunogold labeling experiments have consistently demonstrated the exclusive localization of GARPs in bovine and murine retinas to regions proximal to the rims of membrane discs, similar to the localization of RDS [62]. Further studies suggest that GARP interacts with RDS at the periphery of the discs, thereby mediating the connection between the disc periphery and the plasma membrane and facilitating the stacking of adjacent membrane discs [60,63].

Prominin1 (PROM1) is a pentaspan membrane protein that binds to cholesterol [64] (Figure 1E). In *Xenopus* rod photoreceptor cells, PROM1 is exclusively localized to the base of the OS, while in cones, PROM1 is distributed throughout the entire length of the OS [65]. PROM1 and RDS distribute in a mutually exclusive manner in nascent disc membranes, with PROM1 exclusively present at the “U-shaped” edges, whereas RDS is exclusively localized to the “hairpin-shaped” rims [65]. Expressing the mutant Arg373Cys form of PROM1 protein in mice rod cells can reduces the levels of endogenous wild-type PROM1 protein, disrupts OS disc morphology in photoreceptor cells, and induces gradual apoptosis [66]. Recent studies in zebrafish have revealed that the loss of Prom1b primarily affects the morphogenesis of cone OSs. Additionally, the absence of Prom1b results in the mislocalization of RDS and disrupts its oligomerization [67]. Similar to PROM1, Protocadherin21(PCDH21) also locates at the base of the rod OS and distributes along the edges of open discs [66,68]. PCDH21 can interact with PROM1 and the distribution of PROM1 switches from the base (wild-type) to the entire length of the rod OS in *PCDH21^−/−^* mice mutants [66]. Burgoyne and colleagues discovered through 3D electron tomography that there is a “fiber” ultrastructure present at the newly forming disc edges and adjacent plasma membrane of the IS in mouse rod cells. Further immunogold labeling studies identified this “fiber” as PCDH21 [15]. Consequently, *PCDH21* knockout mice manifest phenotypic traits such as shortened, malformed OSs, and membranous whirls [69]. 

RP1 is a photoreceptor-specific cytoplasmic protein essential for the normal morphogenesis of the OS [70] (Figure 1E). As a member of the doublecortin family, the N-terminus of RP1 contains a tubulin-binding doublecortin (DCX) domain, which binds to tubulin and associates with the photoreceptor axoneme. Immunogold labeling of RP1 in the mouse retina demonstrates that this protein is distributed along the photoreceptor OS axoneme [71]. RP1 promotes microtubule polymerization and enhances its stability. The deletion of the DCX domain shortens the length of the photoreceptor OS axoneme in mice [71]. Mutations in the *RP1* gene can lead to both autosomal recessive and dominant retinitis pigmentosa [72]. It is estimated that 3–4% of RP patients are affected by the loss of RP1 function [73].

The USH1 protein complex is associated with calyceal processes (CP), which are enigmatic microvilli surrounding the base of photoreceptor OSs [74]. Microvilli are primarily composed of F-actin [75]. Investigations have unveiled abnormal morphologies in photoreceptor OSs of *Xenopus tropicalis* following the knockdown of Protocadherin-15 and the knockout of Cadherin-23 (Cdh23) (USH1D protein), characterized by excessive rod basal disc growth and cone OS bending [76]. Zebrafish photoreceptors have similar CPs [77]. In zebrafish, different mutations in USH1 proteins result in varying phenotypes in photoreceptor cells. Protocadherin-15 is expressed in the CPs, OS, and synapses of zebrafish photoreceptors. Its absence leads to abnormal development of photoreceptor OSs, exhibit abnormal directional growth of OS discs. Although Cdh23 is expressed in the zebrafish retina, it is not detected in photoreceptors and *cdh23* mutants do not exhibit defects in photoreceptor morphology and function [78]. Overall, USH1D proteins are crucial for controlling OS disc size and mechanical support in photoreceptor cells.

## 3. Photoreceptor OS Renewal

### 3.1. The Discovery of OS Renewal

Photoreceptor OSs are dynamic structures undergoing continuous renewal to ensure the sustained viability of photoreceptor cells. In 1967, Young’s laboratory used autoradiography to discover that the discs in rod OSs of mice are continuously renewed. Initially, these labels were localized to the IS, forming a distinct bright band at the base of the OS. Over time, this radioactive signal gradually migrated towards the apex of the OS, eventually dissipating from the segment and localizing within the RPE cells. The renewal cycle of OS discs in higher vertebrate rod cells typically spans approximately 10 days [79,80,81] (Figure 2A). Young hypothesized that the uppermost discs undergo active shedding before being phagocytosed by RPE cells. However, subsequent in vitro investigations challenged this hypothesis, revealing that the absence of the RPE in *Xenopus* retina prevents the shedding of the uppermost OS discs. This finding underscores the essential interaction between OS discs and RPE cells for effective shedding [82].

Young also conducted autoradiography experiments to investigate the distribution of radioactive amino acids in the cone OSs of *Xenopus*. However, due to the structural differences between rod and cone OSs, where most cone OS discs are continuous with the plasma membrane, newly synthesized radioactive proteins quickly fill the entire cone OS within a short time [83,84] (Figure 2B). Therefore, at that time, Young believed that cone OSs did not undergo disc renewal as observed in rod OSs, but rather underwent partial molecular turnover. However, in 1977, Steinberg discovered a renewal process similar to rod OSs in cone OSs using squirrels and rhesus monkeys. Steinberg observed that the apex discs of cone OSs were engulfed by pseudopodia extended from RPE cells, and new discs were formed at the base of the OSs [12]. Nonetheless, the exact renewal cycle of cone OSs remains unclear.

### 3.2. Regulation of OS Renewal by Light and Circadian Rhythms

The renewal of rod OSs is highly synchronized with circadian rhythm. After exposure to light in the morning, a large number of apical OSs are phagocytosed by RPE cells, with phagosomes being degraded later in the day [85,86]. In contrast to the shedding of rod OSs, the timing of cone OS shedding varies among different species. However, current research has shown that in almost all species, cone OSs undergo renewal during the night [87].

The circadian rhythm represents an intrinsic biological rhythm governed by alternating light–dark cycles. Thus, the question arises: is the turnover of vertebrate photoreceptor OSs predominantly influenced by external light cues or by intrinsic circadian rhythm? Typically, RPE phagocytosis of rod OS fragments occurs in a burst of activity shortly after daily light onset, while prolonged darkness only marginally alters the peak activity time for rod OS renewal, suggesting that this process is primarily orchestrated by the endogenous circadian rhythm [86]. Furthermore, circadian-regulated renewal of rod OSs can persist for approximately 12 days in mice, underscoring the dominant role of the circadian rhythm in this phenomenon [88]. In contrast to rodents, the renewal of rod OSs in the frog is not regulated by internal circadian rhythm mechanisms but is entirely controlled by light. When frogs are kept in complete darkness for an extended period, the shedding of rod OSs decreases. However, upon exposure to light, there is a significant phagosome peak in RPE. Thus, in frog retinas, the renewal of rod OSs is directly related to light stimulation [85]. 

The circadian rhythm is coordinated and controlled by a set of core clock genes (*Clock, Bmal1, Period,* and *Cryptochrome*). Under the regulation of the endogenous circadian clock, the visual system exhibits a strong circadian rhythm [89,90]. In mammals, the primary biological clock resides in the suprachiasmatic nucleus of the hypothalamus, but additional clocks are present in other brain regions [91]. In lower vertebrates, such as birds, newts, and zebrafish, the pineal gland acts as a circadian pacemaker, regulating behavior by rhythmically synthesizing and releasing melatonin [90,92]. Studies using mice that cannot synthesize melatonin have shown that OS renewal still occurs, but the peak of phagosomes is reduced [93]. *Bmal1, Per1,* and *Per2* knockout in RPE disrupted the daily rhythm of phagosome peaks under cyclic lighting, resulting in a constant number of phagosomes throughout the day, unlike the burst peaks observed in wild-type mice [94,95]. In conclusion, light and circadian rhythms exert pivotal influences on regulating the renewal of photoreceptor OSs.

### 3.3. Molecular Mechanisms of Photoreceptor OS Renewal

In this section, we will elucidate the molecular mechanisms underlying the phagocytosis of OSs by RPE cells. The phagocytic process encompasses the recognition and binding of OSs by RPE, followed by phagocytosis and internalization, and eventual digestion [96].

#### 3.3.1. Recognition and Binding of RPE to OSs 

Recognition and binding represent the initial pivotal steps preceding phagocytosis (Figure 3A). To initiate phagocytosis of the OS apical, the RPE must first recognize them. Under normal conditions, phosphatidylserine (PS) resides on the inner leaflet of the plasma membrane. During apoptosis, PS is translocated to the outer leaflet by the action of “flippases” allowing it to be recognized by receptors on phagocytic cells, thus acting as an “eat me” signal [97,98]. Utilizing the PS-specific probe pSIVA, the apical exposure of PS in the rod OSs can be easily visualized, which suggests a circadian rhythm, especially following the onset of light [99]. Upon recognition of externalized PS, RPE cells typically engage with a ligand-receptor mechanism to bind to the apical of the OS. It remains to be explored whether cone OS renewal also entails a comparable “eat me” signal akin to that identified in rod OSs.

The αvβ5 integrin receptor facilitates the binding of OSs to the surface of RPE cells [100]. Localized specifically to the apical microvilli of the RPE, αvβ5 maintains continuous contact between the RPE and the apical OS. This integrin interacts with PS through the bridging molecule milk fat globule-EGF8 (MFG-E8) (Figure 3A). In mice deficient in either αvβ5 or its ligand MFG-E8, the circadian rhythm of PS externalization is disrupted, and the burst peaks in RPE phagocytosis following light exposure does not occur. Instead, these mice exhibit a consistent basal level of RPE phagocytosis throughout the day [101]. Furthermore, *β5^−/−^* mice experience a significant decline in retinal light response as they age, underscoring the critical role of the αvβ5 integrin receptor in the synchronized phagocytosis of the retina [100].

ANXA5 is another molecule that regulates the binding of OSs to the surface of RPE cells (Figure 3A). Similar to the retinal phenotype observed in *β5^−/−^* mice, *Anxa5^−/−^* mice exhibit a loss of circadian rhythm in the externalization of PS within the RPE. Moreover, the absence of ANXA5 protein reduces the levels of αvβ5 integrin receptors on the apical phagocytic surface of RPE cells [102].

#### 3.3.2. Phagocytosis and Internalization 

Following the recognition and binding of the RPE pseudopodia to the apical of the OSs, the process of phagocytosis is initiated. This process requires cytoskeletal rearrangement to form the phagocytic cup [103]. Two primary signaling pathways mediate this process (Figure 3B). The first pathway involves the MER Proto-Oncogene, Tyrosine Kinase (MerTK), which regulates F-actin through the small GTPase Rac1 [96,103,104]. The second pathway primarily relies on the activation of focal adhesion kinase (FAK). FAK is a multi-domain cytoplasmic tyrosine kinase involved in integrin-mediated outside-in signaling in various physiological contexts. Phosphorylation of FAK at the Tyr861 site enhances its binding to the αvβ5 integrin [105,106,107,108]. Additionally, inhibition of FAK signaling results in decreased MerTK phosphorylation [105].

MerTK serves as a pivotal receptor orchestrating the formation of the phagocytic cup through the apical microvilli of RPE, facilitating the engulfment and phagocytosis of photoreceptor OSs [109]. It is a member of the Tyro–Axl–Mer (TAM) receptor family and localizes within the microvilli of RPE cells. Initially identified in Royal College of Surgeons (RCS) rats, mutations in the *MerTK* gene result in the failure of RPE cells to phagocytose the apical OSs. Consequently, this phagocytic impairment leads to the inability to renew OS discs, ultimately causing structural defects and complete loss of OSs within three months [110,111]. Bridging molecules such as vitamin K-dependent factor protein S (ProS) and growth arrest-specific gene 6 (Gas6) play essential roles in linking MerTK to PS on the OSs [96,112]. Upon binding with its ligands Gas6 or protein S, MerTK facilitates the recruitment of several SH2-domain-containing proteins to phosphorylated tyrosine residues. Furthermore, MerTK can activate phosphoinositide 3-kinase (PI3K), which aids in alleviating membrane tension by dismantling linear actin networks [103]. Additionally, in vitro investigations have demonstrated that MerTK can induce Rac activation by recruiting the p130Cas/CrkII/Dock180 GEF complex. Subsequently, Rac activation triggers the activation of the WASP-family verprolin-homologous protein (WAVE), facilitating the transport of actin monomers to the actin nucleating complex Arp2/3 and promoting branched actin polymerization [103,113] (Figure 3B).

AnnexinA2 (ANXA2) is a membrane-associated protein that regulates the dynamic changes of actin. ANXA2 localizes to the phagocytic cup and early phagosomes, where it may function upstream of focal adhesion kinase (FAK). *Anxa2^−/−^* mice exhibit delayed phosphorylation of FAK following light exposure and display phenotypes characterized by the accumulation of phagosomes in apical microvilli [82,114].

The culmination of RPE phagocytosis involves the closure of newly formed phagosomes at the apical of pseudopodia, a process that remains relatively poorly understood. Phagocytic cup closure typically relies on myosin to generate the requisite force, retracting the elongated pseudopodia and drawing the engulfed OS into the RPE cell [115]. Investigations have unveiled an interaction between MyosinII and MerTk during the engulfment of OSs [116]. Nonetheless, the precise mechanism governing the inward contraction force required to finalize the phagocytic process remains obscure.

#### 3.3.3. Degradation of Phagosomes in RPE Cells

The final steps of OS phagocytosis involve the formation and degradation of phagosomes (Figure 3C). Phagosome maturation is a complex process that involves multiple fusion events, including fusion with early endosomes, late endosomes, and lysosomes. The resultant structure after fusion with lysosomes is termed as the phagolysosome. Each stage of maturation is characterized by alterations in both membrane composition and luminal protein content, accompanied by a gradual acidification of the lumen and an increase in active oxygen [103]. The acidification of the phagosome lumen is primarily mediated by the activity of vacuolar ATPase (V-ATPase). Numerous digestive hydrolytic enzymes are delivered to the phagosomes through fusion with lysosomes, where they undergo activation triggered by a decrease in pH [117,118]. Proper phagosome acidification is essential for the efficient digestion of photoreceptor OSs. Experimental studies have demonstrated that inhibition of V-ATPase in the RPE results in the accumulation of phagolysosomes incapable of degrading opsin [117,119]. The soluble lysosomal aspartic protease cathepsin D plays a vital role in the degradation of visual proteins and rhodopsin within the phagosomes [120]. The Rab family of small GTPases is also essential for phagosome maturation, with Rab7 specifically facilitating the fusion of phagosomes with late endosomes and lysosomes and coordinating the inward movement of phagosomes [103].

Mutation in the *MYO7A* results in Usher syndrome type 1B (USH1B), the predominant form of Usher syndrome, a condition characterized by deafness and blindness. Studies have revealed that in the retinas of *Myo7a* mutant mice, the RPE displays aberrant phagocytosis of OSs, characterized by slow phagosome maturation, and mislocalization of opsins in photoreceptor cells. The deficiency of MYO7A results in a decelerated transport of engulfed OS fragments within the RPE towards the basal region [121]. Investigations have also highlighted the involvement of MYO6 in phagosome clearance, where its downregulation leads to a delay in the degradation rate of phagosomes [122].

Ataxin-3 (ATXN3) also plays a crucial role in phagosome degradation. The loss of ATXN3 in mice results in a phenotype similar to that of *Myo7a* mutant mice. In *Atxn3^−/−^* mice, the number of phagosomes increases in the apical region of RPE cells, while the number decreases in the basal region, indicating a delay in the transport of phagosomes from the apical to the basal region [123]. Similarly, the loss of Atxn3 protein in zebrafish leads to a delay in the transport of phagosomes from the apical to the basal region and the mislocalization of opsin in cone cells [123].

#### 3.3.4. Regulation of OS Renewal by Proteins in Photoreceptor Cells

The components of photoreceptor cells also contribute to the process of OS renewal. For instance, RDS is distributed rhythmically, alternating with the distribution of rhodopsin, within the rodent rod OS [124]. It has been proposed that regions enriched with RDS may serve as breakpoints at the apical of the OSs [124]. However, heterozygous RDS mutant mice, with RDS levels halved, exhibit a higher number of phagosomes within the RPE compared to wild types, thereby challenging this hypothesis [52]. In a recent investigation focusing on rhodopsin transport to OSs, rhythmic layering of rhodopsin within the rod OSs of mice was observed, a phenomenon influenced by both light exposure and circadian rhythms [125]. The authors proposed that the diminished levels of rhodopsin are more susceptible to breakage of OSs [125].

Kif17 is a ciliary motor protein participating in the transport of OS components [126]. Interestingly, phosphorylation of Kif17 also promotes membrane shedding in cone photoreceptor cells. Absence of Kif17 results in defective shedding of discs, while transient expression of phosphorylation-mimicking Kif17(S815D)-GFP in cone cells induces a notable increase in phagosome numbers [127]. RAB28 is another crucial player in ciliary transport and functions as a molecular switch across diverse biological pathways and processes [128]. In the absence of Rab28, phagocytosis is significantly diminished both in rod and cone OSs [129,130,131]. For instance, cone phagosomes were almost absent in *Rab28^−/−^* mice [131]. Zebrafish mutants deficient in Rab28 exhibit approximately a 50% reduction in phagosome numbers [129,130]. These findings underscore the intricate interplay of various proteins and signaling pathways in governing the renewal of photoreceptor OSs.

### 3.4. Retinal Diseases Related with RPE Phagocytic Defects

Given the pivotal role of the RPE in the renewal process of photoreceptor OSs, dysfunction in the phagocytic capacity of RPE cells can lead to structural and functional abnormalities of the OS, resulting in the development of various retinal diseases. 

*MERTK* gene mutations were identified shortly after the discovery of defects in photoreceptor OS renewal-related retinal diseases. These mutations in *MERTK* account for approximately 1.7% of cases of IRD with a severe retinal phenotype [132]. Deletion of this gene affects the phagocytic ability of the RPE, leading to the accumulation of disordered fragments between the apical of the OSs and the pigment epithelium, ultimately resulting in the death of photoreceptor cells [133].

*CERKL* is also a known gene implicated in RP and was first identified in a recessive RP Spanish family in 2004 [134]. Mutations in CERKL lead to autosomal recessive cone-rod dystrophy (arCORD) with inner retinopathy [135]. Zebrafish *cerkl* mutants exhibited progressive degeneration of OSs and increased apoptosis of retinal cells, including those in the outer and inner retinal layers. The lack of Cerkl results in downregulation of MerTK, thereby affecting the phagocytic function of RPE towards photoreceptor OSs [135,136].

RAB28 was identified as the first Rab protein associated with Cone-rod dystrophies (CORD). In humans, null and hypomorphic alleles of RAB28 lead to arCORD. Patients often manifest excessive macular pigment deposition, progressive loss of vision, and atrophy of the RPE, potentially resulting from defects in the RPE cell’s ability to phagocytose OSs [137].

## 4. Prospects

The OSs of photoreceptor cells are crucial in the process of vision formation. However, our understanding of the structure of photoreceptor OSs remains limited. In recent years, the rapid development of cellular imaging technologies has provided new insights. Cryo-electron tomography (cryo-ET) is an emerging technique that offers the advantage of preserving sample integrity more effectively without chemical fixation or heavy-metal stains, making it well-suited for studying the detailed ultrastructure of OSs [138,139]. Superresolution fluorescence imaging techniques allow for more precise protein localization, while Expansion Microscopy (ExM) enables fluorescence-based imaging at nanoscale resolution on standard microscopes [138,140]. Applying these techniques to the study of the ultrastructure of photoreceptor OSs will deepen our understanding of these structures. The proper formation and maintenance of outer segment structure depend on the coordinated action of numerous proteins, such as rhodopsin, RDS, and CNG channels. Yet, many proteins involved in this process remain to be explored. The advancement of CRISPR/Cas9 technology will further facilitate the study of photoreceptor protein functions.

Photoreceptor OSs are not static; they undergo rhythmic phagocytosis by RPE cells. The phagocytic function of RPE cells is essential for the health of photoreceptor cells and our vision. RPE cells engulf the apex of the OSs, requiring the coordinated action of many proteins. Currently, the detailed mechanisms underlying rod OS renewal remain to be investigated, especially in regards to how the RPE cells and OS interact with each other. Our vision mainly depends on the cone photoreceptors, although, unfortunately, much less is known about how cone OSs are renewed. Notably, rodents are nocturnal animals and usually contain a small percentage of cones, resulting a challenge for the mechanism study of cone OS renewal. In this way, investigation of diurnal animals with a large population of cone photoreceptors may provide an alternative model. Zebrafish can be a good model to investigate cone photoreceptors, as in zebrafish cones account for approximately 60% of photoreceptors, compared to only 3% in mice [141]. Moreover, zebrafish undergo rapid visual system development, with larvae exhibiting visual functionality as early as 5 days old, in contrast to mice, which require approximately 15–20 days [142]. Future studies using zebrafish models may facilitate our understanding of the molecular mechanisms underlying cone OS renewal.

Finally, future research should focus on developing therapies for genetic diseases related to retinal defects. Among the organs affected by genetic diseases, the eye is particularly well-suited for gene and cell transplantation therapies due to its highly compartmentalized structure and degree of immune privilege [143]. Advances in engineered viral vectors and transgenic promoters have enabled gene therapy to selectively target different types of retinal cells. For instance, initial Phase I clinical trials showed that subretinal injection of rAAV2-VMD2-*hMERTK* into six patients resulted in no significant adverse effects, and three patients experiencing notable improvements in vision, although the improvements diminished after two years [144]. Additionally, the CRISPR/Cas9 genome editing system represents a novel gene therapy strategy, with ongoing improvements enhancing its efficacy and precision [145]. Despite the promise of these gene therapy techniques, significant challenges remain. Many retinal diseases are complex disorders caused by numerous pathogenic genes, making the identification of these genes a critical task. Additionally, delivery vectors also have limited carrying capacity, complicating the transport of the complete genome editing system. Safety concerns, such as off-target editing and potential tumorigenesis resulting from unintended effects on oncogenes or tumor suppressor genes, also need to be addressed to advance these therapies [145,146]. Advancements in cell biology have enabled the direct reprogramming of human somatic cells into induced pluripotent stem cells (iPSCs), which can then be differentiated into specific human cell types or tissues. This development facilitates the study of photoreceptor cell biology and dysfunction through retinal organoid culture systems. Notably, utilizing stem cells derived from pathological donors allows researchers to investigate the defects associated with inherited retinal diseases, offering potential avenues for treatment [147]. Continued research and innovation in these areas are essential for overcoming these obstacles and realizing the full potential of gene and cell therapies for retinal genetic diseases.

## Figures and Tables

**Figure 1 cells-13-01357-f001:**
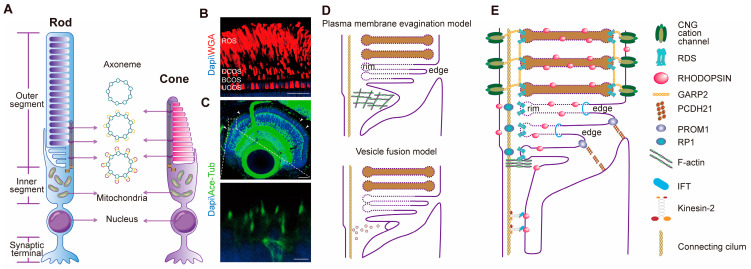
Diagram illustrating the morphogenesis and molecular composition of photoreceptor OSs. (**A**) Schematic diagram of rod and cone structures. The distribution of OSs, inner segments, nuclei, and synapses in rods and cones. (**B**) Wheat germ agglutinin (WGA) labeling of all photoreceptor outer segments in adult zebrafish. ROS: rod outer segment. DCOS: double cone outer segment. BCOS: blue cone outer segment. UCOS: UV cone outer segment. Scale bar 25 μm. (**C**) Anti-acetylated tubulin labeling of photoreceptor connecting cilia in 5 dpf zebrafish. dpf: days post-fertilization. Zebrafish eye immunofluorescence image scale bar: 25 μm. Boxed area magnified image scale bar: 2.5 μm. (**D**) Two modes of rod OS formation. The plasma membrane evagination model proposes that the plasma membrane at the base of the rod cell OS evaginates under the action of F-actin, forming new membranous discs. The vesicle fusion model suggests that new membranous discs are formed by the fusion of vesicles at the base of the OS. (**E**) Molecular structure diagram of rod OSs.

**Figure 2 cells-13-01357-f002:**
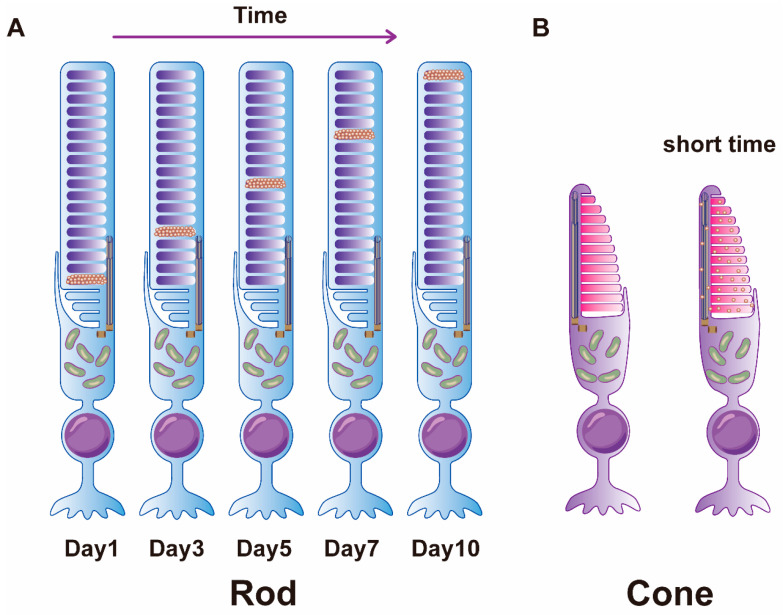
Diagram of photoreceptor OSs renewal. (**A**) Diagram of rod cell OS renewal. Newly synthesized rhodopsin continuously migrates to the apical of the rod OS. (**B**) Diagram of cone cell OS renewal. Newly synthesized cone opsin quickly fills the entire cone OS.

**Figure 3 cells-13-01357-f003:**
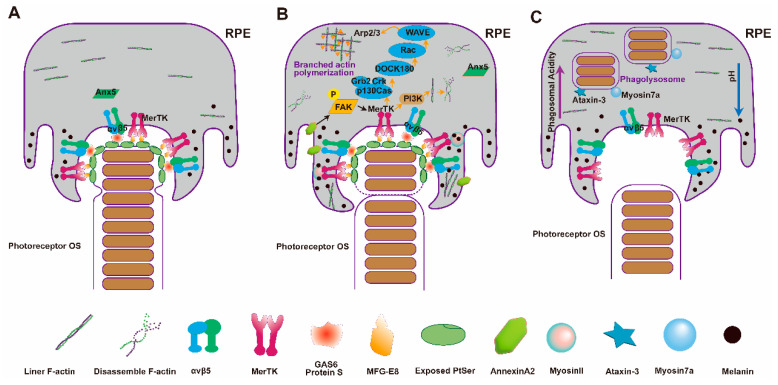
Molecular patterns of RPE cell phagocytosis of photoreceptor cells. (**A**) Pattern diagram of RPE recognition and binding with photoreceptor OSs. At the apical of rod cell OS, phosphatidylserine flips from the inner to the outer leaflet of the membrane and binds with the αvβ5 receptor. (**B**) Pattern diagram of RPE phagocytosis and internalization with photoreceptor OSs. The MerTK receptor regulates the cytoskeleton to form phagocytic cups. (**C**) Pattern diagram of RPE degrading phagosome. Under the action of Ataxin-3 and Myosin7a, phagosomes move from the apical membrane to the basal membrane in RPE cells, where they form phagolysosomes and are degraded.

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
