# Peer review of "The Formation and Renewal of Photoreceptor Outer Segments"

_cells, 2024, doi:10.3390/cells13161357_

Round 1

Reviewer 1 Report

Comments and Suggestions for Authors

Attached

Comments on the Quality of English Language

As mentioned above, these quality of English language is, on the whole good. There are minor spelling and grammar issues that need to be addressed to improve the manuscript but this will not be a long or difficult process.

Reviewer 2 Report

Comments and Suggestions for Authors

The MS by Xu and others is a comprehensive review dealing with major aspects of the pathophysiology of photoreceptor outer segments. It is a timely and adequate review that revises thoroughly the literature on this specific topic. Minor English typos should be corrected. 

Comments on the Quality of English Language

A mentioned to the authors, the MS by Xu and others is a comprehensive review dealing with major aspects of the pathophysiology of photoreceptor outer segments. It is a timely and adequate review that revises thoroughly the literature on this specific topic. Minor English typos should be corrected.

Reviewer 3 Report

Comments and Suggestions for Authors

This nice review is dedicated to a highly specialized region of photoreceptors, the outer segment (OS), represented by the apical part of these cells, deputed to light capturing and phototransduction. Particular emphasis is given to the fact that outer segments are highly dinamic and require rhythmic renewal to maintain normal physiological functions. The review aims to elucidate the structure of photoreceptor OSs, the biological mechanism of photoreceptor OS renewal, and the retinal diseases resulting from defects in this process.

The review is conceived schematically, clealry written and the biological mechanisms explained with sufficient detail.

Suggestions are provided below for a more comprehensive analysis of the intriguing chapter of sensory physiology treated in the manuscript:

1) The text is highly specialistic and abundant in molecule names and chemical notions. To make it easier for readers to follow the flow of the text, it is suggested to insert some illustrations that represents microscopic images of outer segments and cilia, rather than using cartoons alone. The literature shows examples of many beautiful outer segment and cilia which could enrich the biological part of this review, helping memorizing the details.

2) There are highly modern methods to study cilia (and small organelles in general) which have been applied successfully to photoreceptors. Expansion microscopy and super resolution are among these methods, which should be briefly illustrated to give an idea of the state-of-the art approaches available to study cell biology of sensory organs.

Example in the literature are:

Wensel TG, Potter VL, Moye A, Zhang Z, Robichaux MA. Structure and dynamics of photoreceptor sensory cilia. Pflugers Arch. 2021 Sep;473(9):1517-1537. doi: 10.1007/s00424-021-02564-9.

 Moye AR, Robichaux MA, Wensel T. Expansion Microscopy of Mouse Photoreceptor Cilia. Adv Exp Med Biol. 2023;1415:395-402. doi: 10.1007/978-3-031-27681-1_58. 

Wensel TG, Zhang Z, Anastassov IA, Gilliam JC, He F, Schmid MF, Robichaux MA. Structural and molecular bases of rod photoreceptor morphogenesis and disease. Prog Retin Eye Res. 2016 Nov;55:32-51. 

3) Similalry, organoids are used to study photoreceptor biology and dysfunction in a cultured-controlled environment. Although outer segments are quite short in retinal organoids, cilia can be analysed in detail, specifically using stem cells from pathological donors to understand defects occurring in inherited retinal diseases. This approach should be mentioned.  

Kruczek K, Swaroop A. Patient stem cell-derived in vitro disease models for developing novel therapies of retinal ciliopathies. Curr Top Dev Biol. 2023;155:127-163.

It is believed that with these addictions the review will give an account of present research in the field of photoreceptor organization, becoming a solid example of reference literature in this field.

Comments on the Quality of English Language

Language is adequate.
